# A carboxy-terminal ubiquitylation site regulates androgen receptor activity

Seiji Arai [1,2,4], Yanfei Gao[1,3,4], Ziyang Yu[1,4], Lisha Xie[1,4], Liyang Wang[1], Tengfei Zhang[1], Mannan Nouri [1], Shaoyong Chen [1], John M. Asara[1] & Steven P. Balk [1✉]

Degradation of unliganded androgen receptor (AR) in prostate cancer cells can be prevented by proteasome inhibition, but this is associated with only modest increases in polyubiquitylated AR. An inhibitor (VLX1570) of the deubiquitylases associated with the proteasome did not increase ubiquitylation of unliganded AR, indicating that AR is not targeted by these deubiquitylases. We then identified a series of AR ubiquitylation sites, including a not previously identified site at K911, as well as methylation sites and previously identified phosphorylation sites. Mutagenesis of K911 increases AR stability, chromatin binding, and transcriptional activity. We further found that K313, a previously reported ubiquitylation site, could also be methylated and acetylated. Mutagenesis of K313, in combination with K318, increases AR transcriptional activity, indicating that distinct posttranslational modifications at K313 differentially regulate AR activity. Together these studies expand the spectrum of AR posttranslational modifications, and indicate that the K911 site may regulate AR turnover on chromatin.

[1] Department of Medicine and Cancer Center, Beth Israel Deaconess Medical Center, Harvard Medical School, Boston, MA, USA. [2] Present address: Department of Urology, Gunma University Hospital, Maebashi, Gunma, Japan. [3] Present address: School of Basic Medical Sciences, Chongqing Medical University, Chongqing 400016, China. [4] These authors contributed equally: Seiji Arai, Yanfei Gao, Ziyang Yu, Lisha Xie. ✉email: sbalk@bidmc.harvard.edu

Androgen receptor (AR) plays a central role in prostate cancer (PCa) development, and androgen deprivation therapy (ADT) to reduce levels of circulating androgens (testosterone and dihydrotestosterone, DHT) is the standard treatment for metastatic PCa. Androgen binding causes a conformational change in the AR ligand binding domain that stabilizes the protein, drives its dimerization and nuclear translocation, and enhances its interaction with transcriptional coactivator proteins, all resulting in transcriptional activation. The unliganded AR associates with an HSP90 complex that maintains it in a conformation capable of binding ligand[1]. In the absence of ligand, the AR undergoes proteasome-mediated degradation, which appears to be regulated by the HSP90-associated ubiquitin ligase STUB/CHIP[2–4]. However, the specific sites that are targeted by STUB, or by other ubiquitin ligases that have been reported to ubiquitylate AR and drive its degradation (including MDM2, SKP2, SIAH2, SPOP)[5–9], remain to be established.

While lysine 48 (K48) linked polyubiquitylation generally drives degradation, monoubiquitylation or polyubiquitylation with other linkages can mediate an array of distinct functions. In particular, RNF6 has been reported to mediate AR ubiquitylation at K845 and K847 in the ligand binding domain through a distinct linkage, resulting in increased transcriptional activity[10]. Similarly, AR can undergo ubiquitylation at K313 by MDM2 in the N-terminal domain, and while mutation of this site decreases degradation it also impairs AR transcriptional activity[11]. Interestingly, while K313 is in the N-terminal domain, ubiquitylation of this site requires the presence of the ligand binding domain. Mechanistically, mutation of this site results in AR retention on chromatin, but a loss of its association with p300, which may be a basis for the loss of transcriptional activity[11]. In addition to ubiquitylation, the AR may undergo many other posttranslational modifications including phosphorylation, methylation, acetylation, SUMOylation, and ADP-ribosylation[12–15]. We report here on a series of mass spectrometry and functional studies focused on AR ubiquitylation, and identify a not previously reported site in the AR carboxy terminus (K911) that contributes to AR stability and transcriptional activity.

## Results

**AR degradation is associated with increased ubiquitylation.** As shown previously, AR protein is decreased in the absence of androgen, and this decrease can be mitigated by proteasome inhibition, indicating the unliganded AR is subject to disposal via the ubiquitin-proteasome system (Fig. 1a). In VCaP PCa cells expressing very high levels of AR (due to AR gene amplification), treatment with a proteasome inhibitor in androgen depleted medium leads to readily detectable expression of high molecular weight AR species (~250 kD) (Fig. 1b), which are not observed in the presence of DHT (Fig. 1c), consistent with ubiquitylation. AR immunoprecipitation followed by ubiquitin immunoblotting confirmed that the high molecular weight species are ubiquitylated AR, while the series of discrete bands running at about 20 – 60 kD above AR are consistent with sumoylation[16] (Fig. 1d).

AR ubiquitylation is less abundant in cells expressing lower levels of AR (LNCaP and C4-2), but proteasome inhibition nonetheless increases the expression of the unmodified AR migrating at ~110 kD in these cells in androgen depleted medium. One possible basis for the low recovery of ubiquitylated AR after proteasome inhibition, and for the increase in unmodified AR, would be ubiquitin-independent proteasomal degradation, which is mediated by the 11S/PA28 cap on the 20S proteasome that recognizes unfolded proteins[17]. To test this we used siRNA to deplete PSME3/PA28γ/REGγ, which forms the 11S cap, but found this did not prevent the degradation of AR in cells treated with the HSP90 inhibitor geldanamycin (GA) (Supplementary Fig. 1).

Alternatively, AR may undergo deubiquitylation by the deubiquitylases (DUBs) associated with the proteasome (PSMD14, USP14, and UCHL5). Notably, USP14 has previously been reported to mediate AR deubiquitylation[18]. To test this hypothesis we examined the effects of b-AP15, a reported antagonist of both USP14 and UCHL5[19,20]. Treatment of LNCaP cells with b-AP15 in androgen depleted medium decreased AR protein, but did not substantially increase levels of high molecular weight AR, indicating that unliganded AR was not being deubiquitylated by USP14 or UCHL5 (Fig. 2a). Conversely, high molecular weight AR species were increased by b-AP15 when cells were cultured in medium with DHT. We then examined effects of VLX1570, a reported more active antagonist of both USP14 and UCHL5, in C4-2 PCa cells[21,22]. VLX1570 decreased AR in AR antagonist (enzalutamide) or DHT treated cells, with the decrease being more pronounced in the DHT treated cells (Fig. 2b). Moreover, similarly to LNCaP cells, this was associated with a substantial increase in high molecular weight AR only in the DHT-treated cells.

In VCaP PCa cells, VLX1570 also more markedly depleted AR and increased high molecular weight AR in cells treated with DHT (Fig. 2c). VCaP cells also express an AR splice variant (AR-V7) that has lost the ligand binding domain, and is constitutively nuclear and active in the absence of ligand[23]. In contrast to the full-length AR, effects of VLX1570 on AR-V7 were comparable in DHT and enzalutamide treated cells, with a marked decrease in AR-V7 running at ~75 kD and an increase in high molecular weight AR-V7.

Remarkably, the depletion of AR and AR-V7, and generation of slower migrating AR species, could be observed within several minutes of VLX1570 addition, which appeared inconsistent with an accumulation of ubiquitylated AR (Fig. 3a). To test whether the slower migrating species were ubiquitylated AR, we overexpressed Flag-tagged AR in LNCaP cells, treated with DHT and VLX1570, and carried out immunoprecipitation of the Flag-tagged AR followed by immunoblotting for ubiquitin. As expected, ubiquitin immunoblotting of the input showed that VLX1570 markedly increased the overall levels of high-molecular weight ubiquitylated proteins (Fig. 3b, input). Moreover, VLX1570 also caused a marked increase in high molecular weight species precipitated by the anti-Flag antibody that were detected by ubiquitin immunoblotting. To confirm that these high molecular weight species were ubiquitylated proteins, we treated lysates with a deubiquitylase (USP21), which led to complete loss of the ubiquitin signal (Fig. 3b). However, despite this loss of ubiquitin, there was no decrease in the slower migrating AR bands detected by immunoblotting with the Flag antibody (Fig. 3b). This indicated that the AR shift to higher molecular weight was not due to polyubiquitylation.

Notably, a recent study found that VLX1570, in addition to antagonizing USP14 and UCHL5, could induce protein crosslinking[24]. Moreover, a previous study found that related compounds with a bis (arylidene)cyclohexanone scaffold preferentially accumulate in the nucleus[25]. This suggested that the higher molecular weight AR species generated by VLX1570 in DHT treated cells reflected protein crosslinking in the nucleus. Consistent with this hypothesis, cell fractionation showed that these high molecular weight AR species were primarily in the nucleus (Supplementary Fig. 2a), and were in both the soluble nuclear and chromatin fractions (Supplementary Fig. 2b).

Finally, while the greater effects of VLX1570 seen with DHT treated cells may reflect increased nuclear localization, it was possible that the conformational changes in AR mediated by agonist binding may make it a better substrate for the crosslinking activity of VLX1570. To test this we prepared whole cell lysates from DHT or enzalutamide treated cells and exposed them in vitro to VLX1570, and found similar shifts to high molecular weight forms in both cases (Supplementary Fig. 3a). Comparable results

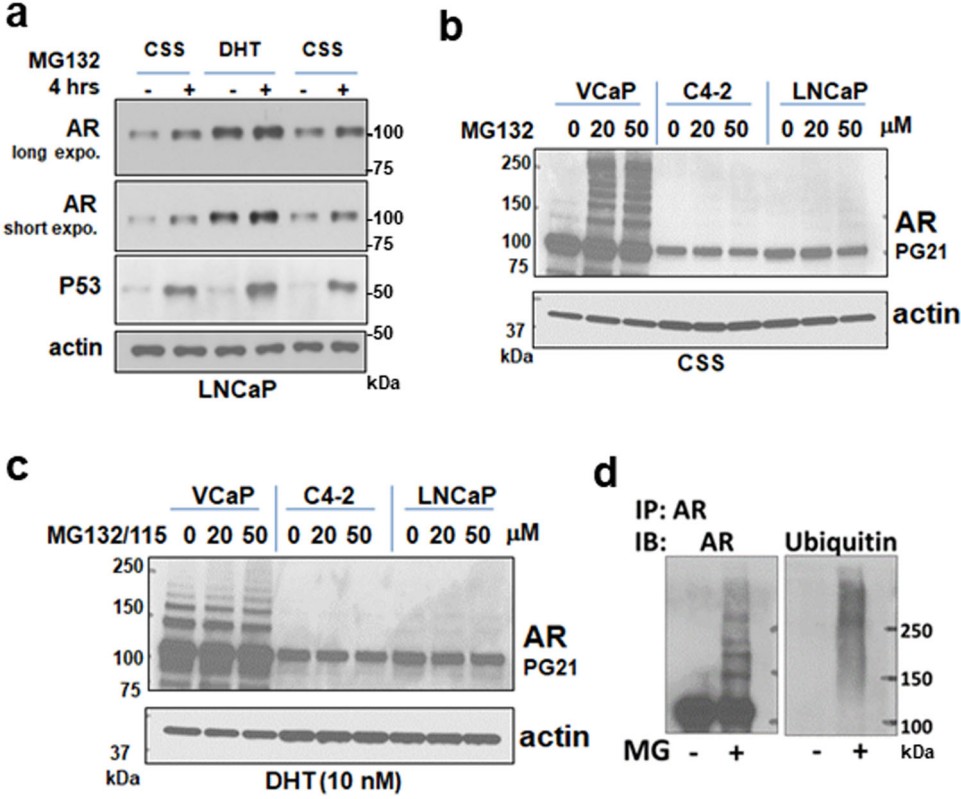

**Fig. 1 Unliganded AR undergoes ubiquitylation and proteasome-mediated degradation. a** LNCaP cells cultured in medium with steroid depleted serum (charcoal stripped serum, CSS) or in CSS with addition of DHT (10 nM for 24 h) were treated with MG132 (20 μM) for 4 h before lysis in RIPA buffer and immunoblotting. **b** VCaP, C4-2, or LNCaP cells in CSS medium were treated with MG132 for 4 h before lysis and immunoblotting with AR N-terminal Ab (PG21). **c** Cells in CSS medium were treated with DHT for 24 h with MG132 for the last 4 h, followed by lysis and immunoblotting. **d** VCaP cells in CSS medium were treated with MG132 for 4 h followed by anti-AR immunoprecipitation and immunoblotting with AR or ubiquitin Ab.

were obtained when we generated nuclear extracts from DHT versus enzalutamide treated cells (Supplementary Fig. 3b). Taken together these data support the conclusion that the marked effects of VLX1570 on the agonist liganded AR reflect protein crosslinking in the nucleus.

**Identification of AR ubiquitylation sites and other post-translational modifications.** As noted above, we were unable to substantially increase levels of ubiquitylated AR by targeting the 11 S cap or 19 S associated DUBs. Nonetheless, we next proceeded with mass spectrometry to identify AR ubiquitylation sites and other PTMs in VCaP cells cultured in steroid depleted medium (+/− DHT) with a proteasome inhibitor (MG132). For these studies we immunopurified AR and excised an AR band from SDS-PAGE at ~110 kD, and excised a region above this band (>125 kD). Ubiquitin sites were found at resides K639, K718, K823, K826, K837, K862, K911 (Table 1, Supplementary Fig. 4). Notably all were associated only with the upper AR band except K911, which was found in upper and lower bands, suggesting some mono-ubiquitination at that site. We also identified multiple ubiquitin derived peptides in the upper band, with the only modification being a glygly remnant at K48, consistent with K48 linked poly-ubiquitylation and targeting for proteasome degradation. Finally, peptides were also identified from SUMO1 or SUMO2, consistent with previous studies showing that AR undergoes sumoylation[16].

We also carried out additional AR mass spectrometry studies with Flag-AR in LNCaP cells and in VCaP cells treated with VLX1570. These confirmed ubiquitylation at K862, and identified an additional site at K313 in high molecular weight AR in VLX1570 treated VCaP cells (Table 1). Previous studies have found AR

ubiquitylation at K313[11], and also at K845 and K847[10], the latter which were not detected here. These latter sites, as well as potentially other sites, may have been missed in our study due to factors including low stoichiometry or incomplete peptide coverage.

We also identified sites of AR phosphorylation, methylation, and acetylation. Serine/threonine phosphorylation sites we found recurrently (S96, S258, S310, S516, S647, S648, T649, T650, and S651) were consistent with previous studies, with S651 being most prevalent (Table 1). One additional recurrent site that has not been previously reported was at S522. Notably, we did not get peptide coverage over S83, which has been previously identified as a major phosphorylation site. We detected tyrosine phosphorylation at Y621 in DHT treated VCaP cells in the nuclear and cytoplasmic fractions in one experiment, and at Y531 in one other experiment in DHT treated VCaP cells (in each case associated with the band at ~110kD) (Table 1). These tyrosine phosphorylation sites have not been described previously.

Lysine methylation (all monomethylation) was found at K313, K318, K634, K639, K659, K823, and K826. Lysine acetylation was found at K222, K313, and K610. Finally, we found arginine methylation (primarily monomethylation) at R13, R20, R101, R129, R264, and R407, with R20 being most consistently detected (Table 1).

**Mutation of ubiquitylation sites decreases AR degradation.** We generated lysine to arginine mutations at identified ubiquitylation sites to assess effects on AR degradation. Mutation of K639, K837, and K911, but not K826, decreased protein degradation after cycloheximide treatment in stably transfected LNCaP cells in androgen depleted medium (Fig. 4a). We next used HSP90 inhibition with geldanamycin to further enhance degradation of

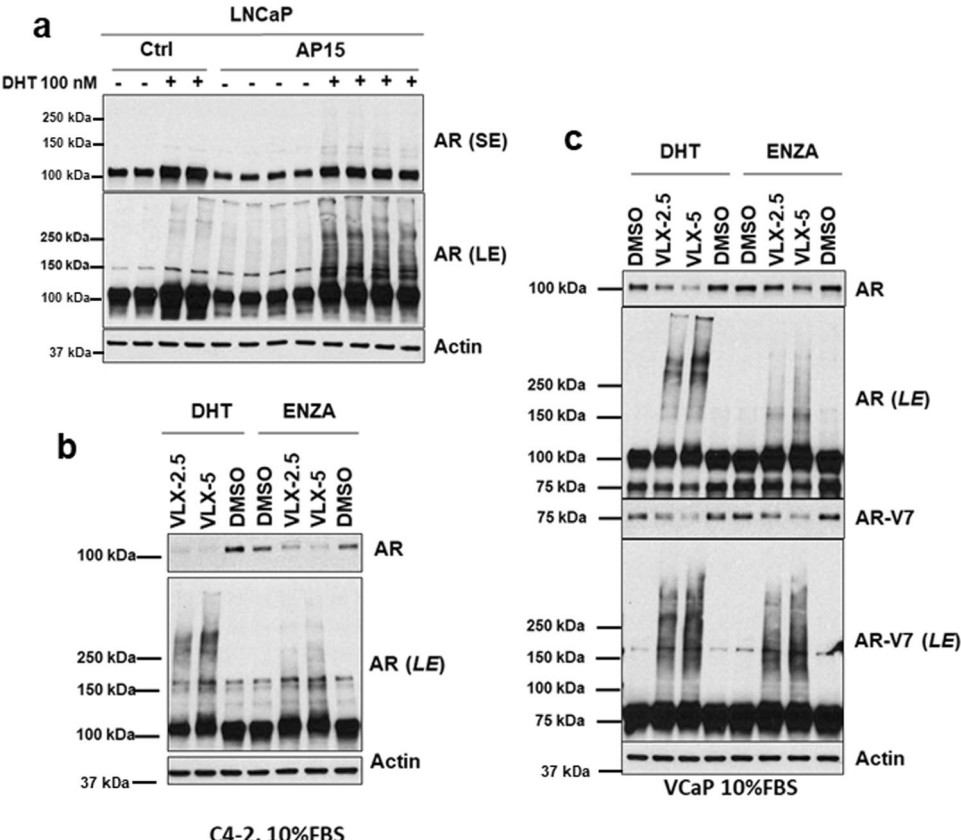

**Fig. 2 Proteasome-associated DUB inhibitors shift DHT-liganded AR to high molecular weight. a** LNCaP cells in CSS medium were treated with vehicle or DHT for 24 h, with addition of b-AP15 (AP15, 5 μM) for last 4 h as indicated. Short exposure (SE) and long exposure (LE) of AR immunoblot with biological replicates is shown. **b** C4-2 cells in medium with 10% FBS were treated for 24 h with DHT (10 nM) or enzalutamide (ENZA, 10 μM), followed by VLX1570 (VLX) at 2.5 or 5.0 μM for last 4 h. **c** VCaP cells in 10% FBS were treated for 24 h with DHT (10 nM) or enzalutamide (10 μM), followed by VLX1570 at 2.5 or 5.0 μM for last 4 h. Lysates were immunoblotted for total AR or with an AR-V7 specific Ab. Short exposures (SE) are shown for the unmodified AR and AR-V7, and long exposures (LE) to visualize high molecular weight species.

wildtype AR, and assess for effects of these mutations. As expected, the stably expressed wildtype AR in PC3 cells was markedly decreased by geldanamycin (Fig. 4b). Geldanamycin also markedly decreased the K826R mutant, while the decrease was less marked for the K911R and K837R mutants. We also transiently expressed the K911R and K837R mutants in C4-2 cells, and again found that the mutants were more stable than the wildtype AR after geldanamycin treatment (Fig. 4c, d). Together these findings indicate that AR degradation may be regulated by ubiquitylation at multiple sites.

**Mutation of K911 enhances transcriptional activity.** We next used an ARE-luciferase reporter to assess the effects of each mutation on AR transcriptional activity, and found that DHT-stimulated luciferase activity was only increased in cells expressing the K911R mutant (Fig. 5a). AR ubiquitylation at K911 has not been characterized previously, and our mass spectrometry data suggested this site may undergo monoubiquitylation as well as poly-ubiquitylation, so we further assessed the functional effects of mutating this site. Consistent with the reporter gene assay, the K911R mutant stably expressed in PC3 cells showed increased DHT stimulated activity relative to the wildtype AR, as assessed by expression of the endogenous *FKBP5* gene (Fig. 5b, c). Notably, AR ChIP-qPCR showed that this increased activity was associated with increased AR binding to the *FKBP5* gene (Fig. 5d). To determine effects on chromatin binding more broadly, we did cell fractionation at 4 and 24 h after DHT stimulation. Levels in the cytoplasm and

soluble nuclear extract were comparable, but the K911R mutant showed greater chromatin binding that was most apparent at 4 h after DHT stimulation (Fig. 5e). Together these results indicate that the turnover of chromatin-associated AR may be increased by K911 ubiquitylation.

**Posttranslational modifications at K313 and K318 decrease AR activity.** K313 was identified previously as a site for AR ubiquitylation (labeled K311 in the previous study). Mutation of this site in that study was found to increase AR stability and chromatin binding, but to impair transcriptional activity due to loss of p300 interaction[11]. In addition to ubiquitylation, we found methylation and acetylation at K313, suggesting that effects of mutating this site may be context dependent. By transient transfection with an ARE-luciferase reporter plasmid in 293 T cells we found comparable activity for the wildtype versus K313A mutant AR (Fig. 6a, b). Notably, our mass spectrometry data also showed K318 methylation, but a K318A mutant similarly had comparable activity to wildtype. In contrast, a double mutant (K313/318A) had increased DHT stimulated activity. This did not appear to be due to differences in AR expression levels as they were comparable for the wildtype and mutant ARs (Fig. 6b). We also generated the double arginine mutant (K313/318 R), which similarly showed comparable levels of expression and increased activity relative to wildtype AR (Fig. 6c, d). We further confirmed the increased activity of the K313/318 A mutant over a broad range of DHT concentrations (Fig. 6e).

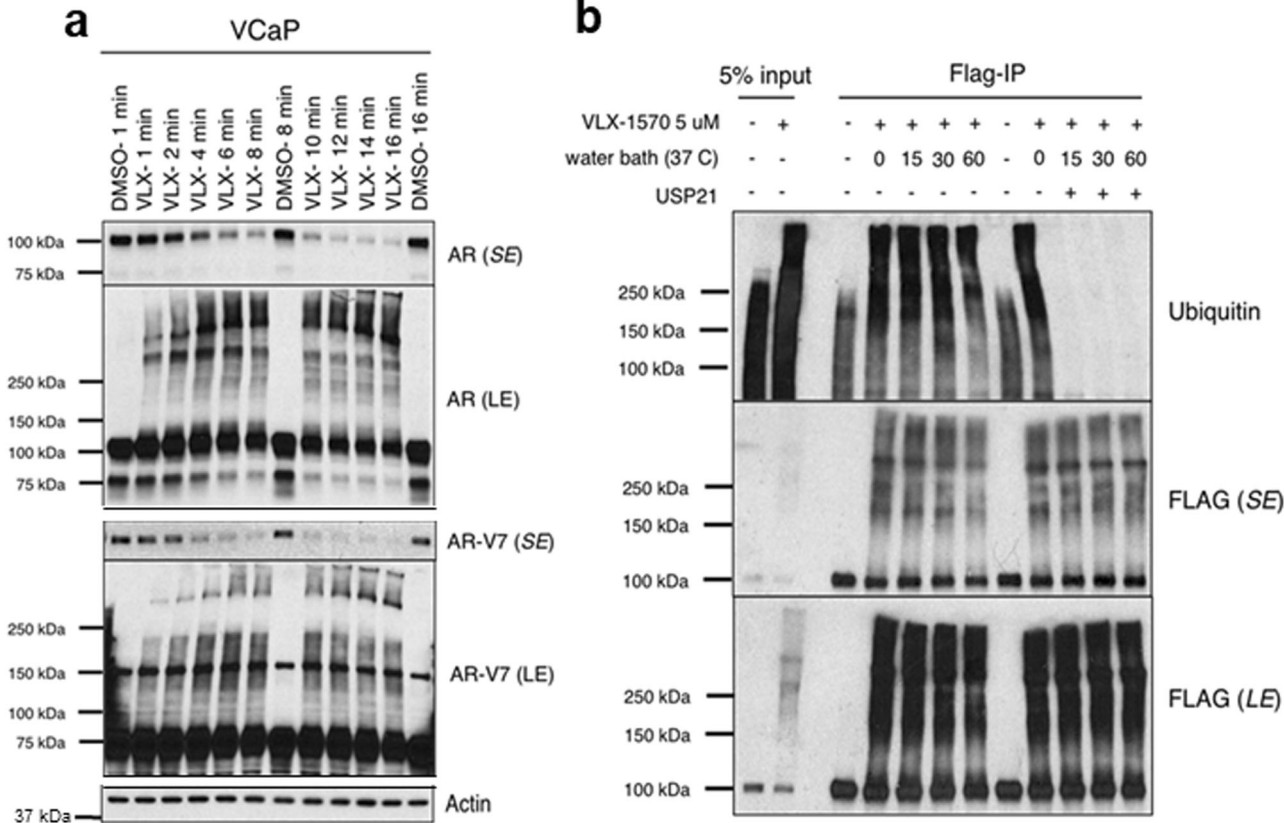

**Fig. 3 High molecular weight AR species generated rapidly by VLX1570 are not ubiquitylated. a** VCaP cells in 10% FBS medium were treated with VLX1570 (5 μM) for 1–16 min followed by lysis in RIPA buffer and immunoblotting for total AR or AR-V7. Long exposures (LE) are shown for the full AR and AR-V7 blots, and above each are shown short exposures (SE) for assessment of the AR and AR-V7 band intensities. **b** LNCaP cells stably expressing an N-terminal Flag-tagged AR were cultured in 5% CSS with addition of DHT (50 nM) for last 2 h, and addition of VLX1570 (5 μM) for the final 15 min before lysis. Lysates were then incubated without/with USP21 (0.5 μM) for 15–60 min, immunoprecipitated with a Flag Ab, and immunoblotted for ubiquitin or Flag. Short (SE) and long (LE) exposures are shown for the Flag tag.

**Table 1 AR posttranslational modifications.**

| Cells | Conditions | | Size | PTMs |
|---|---|---|---|---|
| VCaP | CSS, MG132 | | >125 kD | K837Ub, K911Ub |
| VCaP | CSS, MG132 | | >125 kD | R20me, pS310, pS516, pS522, K634me, pS647, pS648, pT650, pS651, K659me, K823Ub, K826Ub |
| | | | ~110 kD | R13me2, R20me, pS96, pS258, R407me, pS516, K634me, K639me, pS647, pS648, pS651, K659me, K823me, K826ac |
| VCaP | CSS, MG132 | | >125 kD | R129me, pS310, pS516, K639Ub, pS651, K659me |
| | | | ~110 kD | R31me, pS96, R101me, pS121, R129me, pS258, R264me, pS310 |
| VCaP | DHT, MG132 | | >125 kD | R20me, R129me, pS258, pS310, pS651, K718Ub, K862Ub, K911Ub |
| | | | ~110kD | R20me, pS96, K222ac, pS310, K313ac, pS522, K639me, pT650, pS651 |
| VCaP | CSS, MG132 | | >125 kD | R129me, R264me, K634me, K639me, K639Ub, K659me |
| | | | ~110 kD | R31me, R101me, R129me, K313me, K318me, pY531, K659me, K911Ub |
| VCaP | DHT, MG132 | | >125kD | R20me, K718Ub, K862Ub, K911Ub |
| | | | ~110 kD | R20me, K222ac, K313ac, K639me |
| VCaP | DHT, MG132 | cyto | >125 kD | R13me, R20me, pS310, K862Ub |
| | | | ~110 kD | R13me, R20me, pS96, pS518, K619ac, pY621, pS651 |
| | | nuclear | >125 kD | R13me, R20me, pS651 |
| | | | ~110 kD | R13me, R20me, pS96, K619ac, pY621, pS651 |
| LNCaP Flag-AR | CSS, MG132 | | | pS96, pS647, pT650, pS651, K862Ub |
| VCaP | DHT, VLX1570 | | >125 kD | pS96, pS310, K313Ub, pS518, pS651, K659me |
| VCaP | DHT, VLX1570 | | >125 kD | pS96, pT649, pS650, pS651 |
| | | | ~110 kD | pS651 |

Endogenous AR in VCaP cells (or Flag-tagged AR in LNCaP cells) cultured under indicated conditions was immunoprecipitated with an N-terminal AR Ab (or Flag Ab) and run on SDS-PAGE. Areas of the gel at ~110 kD and >125 kD were excised and analyzed by LC-/MS/MS. In one experiment lysates from VCaP cells were separated into nuclear and cytoplasmic fractions prior to ip.

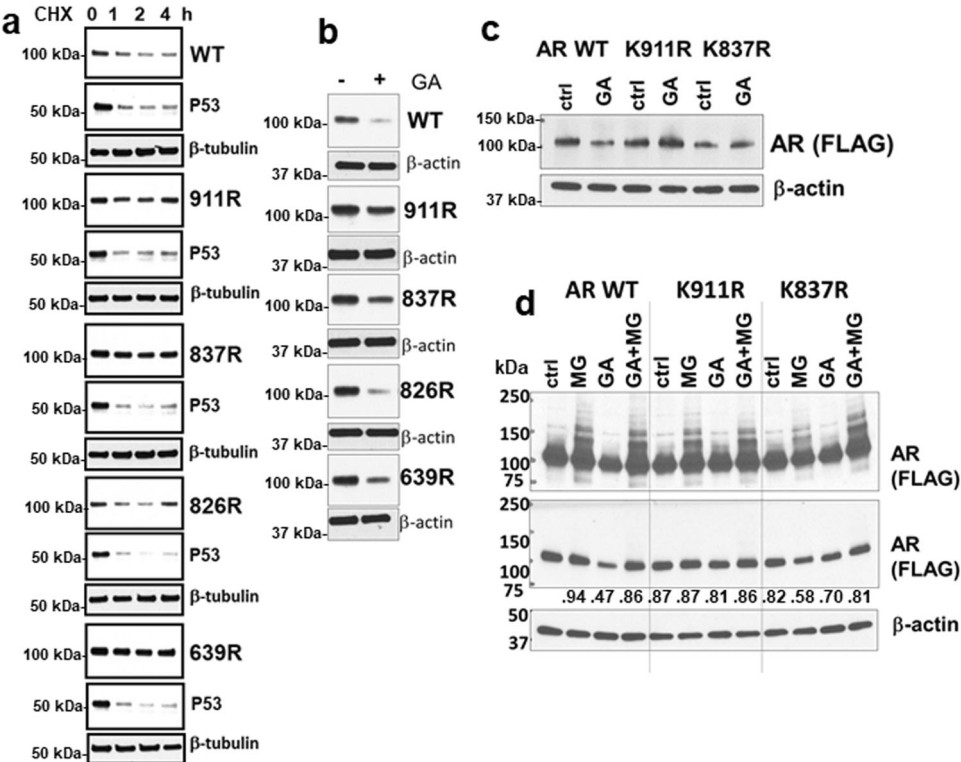

**Fig. 4 Mutation of K639, K837, or K911 decreases AR degradation. a** Wild-type or lysine to arginine AR mutants (V5-tagged) were stably expressed in PC3 cells. Cells were cultured in CSS medium for 24 h, followed by addition of cycloheximide (CHX) for 0 – 4 h. Lysates containing equal amounts of total protein were immunoblotted for AR, p53 (control for effect of CHX), and β-tubulin (protein loading control). **b** PC3 cells stably expressing V5-tagged wild-type or mutant ARs were cultured for 24 h in CSS medium followed by 4 h in geldanamycin (GA). **c** C4-2 cells transiently expressing Flag-tagged AR wild-type or mutants and cultured for 24 hrs in 5% CSS medium were treated with geldanamycin (GA) or vehicle for 4 h and immunoblotted for the Flag tag. **d** C4-2 cells transiently expressing Flag-tagged AR wild-type or mutants and cultured for 24 h in 5% CSS medium were treated with MG132 (MG), geldanamycin (GA), or the combination for 4 h and immunoblotted for the Flag tag. Longer and shorter exposures are shown. Band intensities relative to the AR wildtype (WT) control are quantified below the short exposure.

To assess these mutants in a more physiological context we generated stable transfectants in LNCaP cells, and then used siRNA (targeting the 5'UTR that is not contained in the expression vectors) to deplete the endogenous AR. The ectopic wildtype AR was expressed at slightly higher levels than the mutants, suggesting these sites are not major mediators of AR degradation (Fig. 7a). However, although it was expressed at lower levels, the K313/318 A mutant AR yielded increased DHT-driven expression of the endogenous *KLK3* and *NKX3.1* genes (~50% and ~20% greater than wildtype, respectively). In contrast, activity of the K313/318 R mutant was comparable to the wildtype (Fig. 7b). Due to this lack of enhanced activity we did not further assess K313R or K318R single mutants. To determine whether the mutations were altering AR recruitment to these genes, we carried out AR ChIP-qPCR studies. Consistent with the transcriptional readouts, the K313/318 A mutant showed modestly increased binding relative to the wildtype in response to DHT (Fig. 7c). Interestingly, the K313/318 R mutant had increased basal binding and lower fold induction by DHT (Fig. 7c).

The modest increased activity of the alanine mutant, but not the arginine mutant, suggested the effects may be related to loss of the positive charge rather than decreased methylation, acetylation, or ubiquitylation. We then considered that the effects of mutating K313 or K318 may be more substantial in a context where there is increased activity of methyltransferases (or possibly acetyltransferases or ubiquitin ligases) targeting these sites. To address this we overexpressed a series of lysine methyltransferases that have been implicated as targeting AR, and assessed for effects on wild type

versus the K313/318 A mutant AR. SMYD1 and SETD7 expression both decreased activity of wild type AR, but also decreased the K313/318 A mutant AR (Supplementary Fig. 5a). In contrast, SMYD2 and KMT5A increased activity of the wild type AR. However, activity of the K313/318 A AR was also increased by these methyltransferases. By immunoblotting we confirmed that the methyltransferases had comparable effects on expression of the wild type versus K313/318 A mutant AR (Supplementary Fig. 5b). Finally, we examined whether the stimulatory effects of the lysine demethylase KDM1A/LSD1 may be mediated through these sites. LSD1 increased activity of wild type AR, which may be due in part to increased AR protein (Supplementary Fig. 6). However, the effects of LSD1 on the K313/318 A mutant AR were comparable to the wild type AR, consistent with our previous data showing that LSD1 enhances AR activity through FOXA1 demethylation[26].

## Discussion

Previous studies have established that the unliganded AR has decreased stability and undergoes polyubiquitylation and degradation, with ubiquitin ligases including STUB1/CHIP, MDM2, SPOP, and SKP2 being implicated[2–6,9]. Conversely, AR ubiquitylation by RNF6 can enhance transcriptional activity[10]. The sites targeted by RNF6 are K845/847, but specific ubiquitylation sites driving AR degradation remain to be determined. To map these latter sites we initially attempted to rescue polyubiquitylated AR from proteasome-associated DUBs, but inhibition of USP14 and UCHL5 with b-AP15 or VLX1570 did not increase levels of

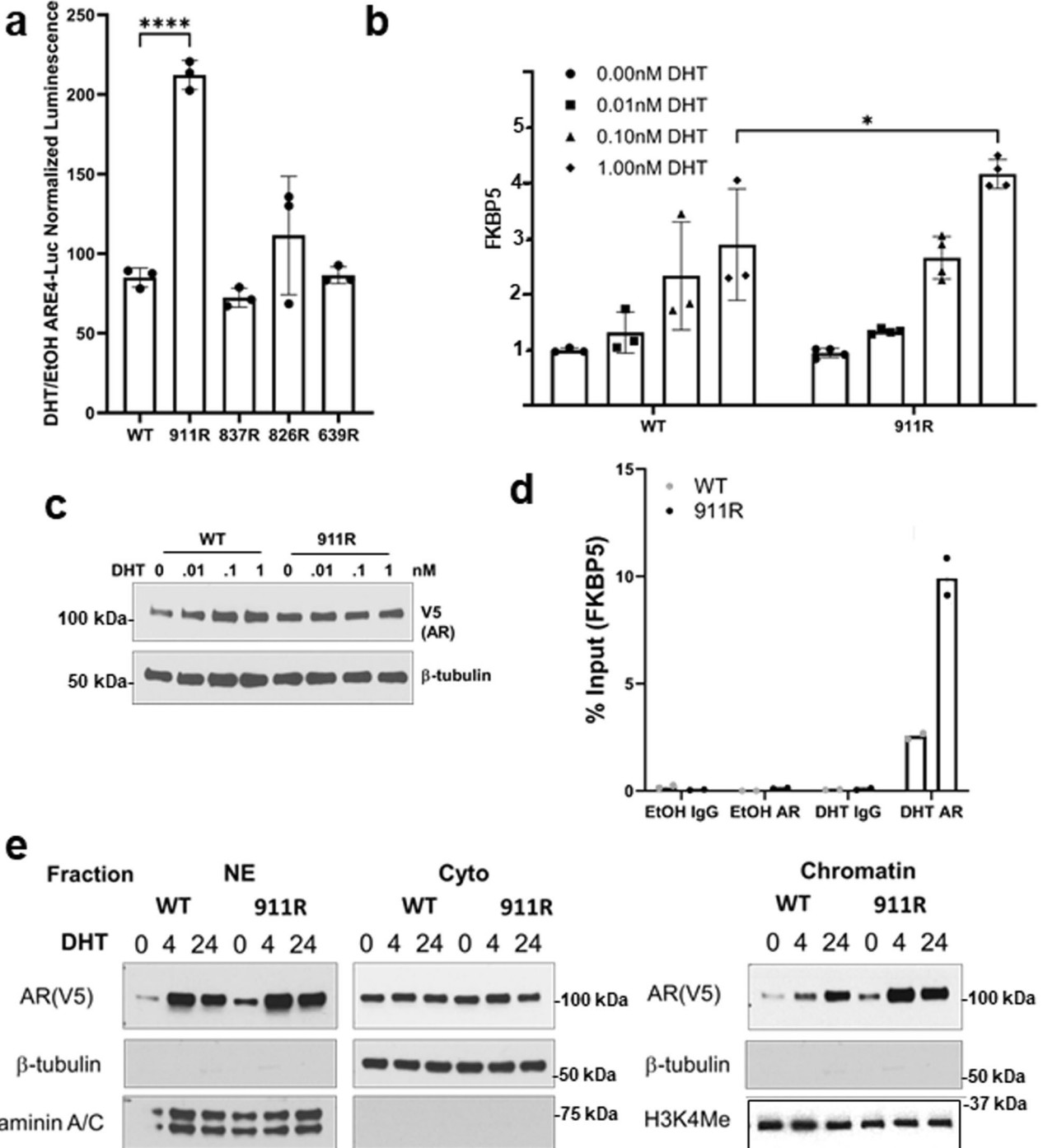

**Fig. 5 AR transcriptional activity is enhanced by K911R mutation. a** PC3 cells stably expressing wild-type or mutant ARs were cultured in CSS medium and transiently transfected with an ARE-luciferase reporter vector for 48 h, followed by 24 h treatment with DHT or vehicle. Fold induction and SEM of luciferase activity in biological replicate samples is shown. (****$p < 0.0001$) **b** PC3 cells stably expressing wild-type or mutant ARs were cultured in CSS medium and treated for 24 h with DHT (0.01 – 1 nM). FKBP5 expression was then assessed by qRT-PCR in biological replicate samples. Fold induction and SEM in biological replicates is shown (*$p < 0.01$). **c** AR protein expression in parallel wells from experiment in (**b**). **d** PC3 cells stably expressing wild-type or K911R mutants ARs were cultured in CSS medium and treated for 4 h with vehicle or DHT (10 nM). AR binding to the ARE-regulated FKBP5 enhancer was then assessed by ChIP-qPCR, and expressed as fold change relative to the IgG vehicle wildtype AR. Binding was normalized to WT IgG in vehicle (ethanol) and mean of biological replicates is shown. **e** PC3 cells stably expressing wild-type or K911R mutants ARs (V5 tagged) were cultured in CSS medium and stimulated with DHT (10 nM) for 0, 4, or 24 h. Cells were then lysed and separated into cytoplasmic (Cyto), soluble nuclear (NE), and chromatin (Chromatin) fractions. Blot is representative of three experiments.

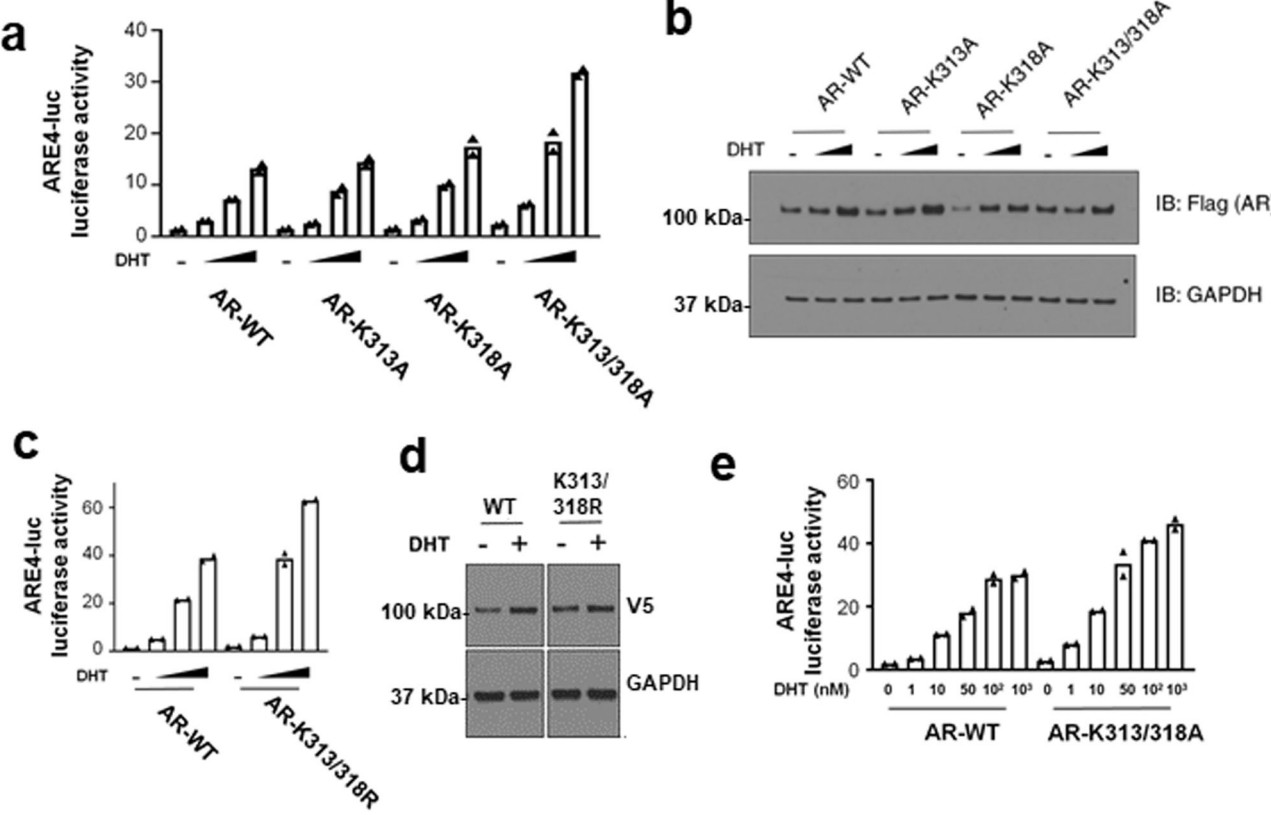

**Fig. 6 Loss of K313 and K318 enhances AR activity on exogenous reporter gene. a** Wild-type or mutant ARs (Flag-tagged) were transiently expressed in 293 T cells with an ARE-luciferase reporter and activity was assessed in response to 0.1, 1, or 10 nM DHT. Data are mean and of biological replicate samples. **b** Flag-AR protein expression under each condition was assessed in parallel with experiment in (**a**). **c** Wild-type or K313/318 R AR (V5-tagged) were transiently expressed in 293 T cells with an ARE-luciferase reporter and activity was assessed in response to 0.1, 1, or 10 nM DHT. Data are mean of biological replicate samples. **d** V5-AR protein expression at 0 and 10 nM DHT was assessed in parallel with the experiment in (**c**). **e** Wild-type or K313/318 A AR were transiently expressed in 293 T cells with an ARE-luciferase reporter and activity was assessed in response 1 – 1000 nM DHT. Data are mean of biological replicate samples.

polyubiquitylated AR. It remains possible that AR deubiquitylation after proteasome inhibition is mediated by the proteasome associated DUB PSMD14, or that proteasome inhibitors instead suppress AR polyubiquitylation by depletion of ubiquitin pools or other mechanisms. Notably, we found that b-AP15 and VLX1570 caused a rapid and dramatic shift of DHT-liganded AR to higher molecular weights, but established that this did not reflect polyubiquitylation. Consistent with these results, VLX1570 may concentrate in the nucleus and has been reported to cause protein crosslinking[24,25]. Notably, the rapid and potent effects of VLX1570 on AR and AR-V7 suggest it may have particular affinity for AR, but further studies are needed to test this hypothesis.

We subsequently carried out mass spectrometry studies in androgen depleted or DHT stimulated cells and identified a series of AR ubiquitylation sites, as well as other PTMs. Ubiquitylation sites at K313, K639, K718, K823, K826, K837, and K862 were all found only in high molecular weight AR species, consistent with polyubiquitylation, while a K911 site was also found in AR migrating between ~110-125 kD, suggesting this site may also be monoubiquitylated. Mutation of K639, K837, or K911 to arginine increased the stability of AR in steroid depleted medium, and in response to HSP90 inhibition with geldanamycin, indicating that polyubiquitylation at multiple sites may drive AR degradation.

The K911R mutation also enhanced DHT-stimulated transcriptional activity. This did not appear due to effects on AR stability as AR protein levels in DHT treated cells were not greater in the K911R expressing cells. Moreover, chromatin fractionation and ChIP-qPCR studies indicated that the K911R mutant had

increased chromatin binding. In the AR LBD crystal structure, K911 is located in the AR C-terminus at a turn separating a short helix and a beta sheet, with the side chain facing out and not having intramolecular contacts, and hence potentially available for ubiquitylation[27]. Notably, a recent study using cryo-EM combined with cross-linking mass spectrometry found that an adjacent lysine (K913) was in proximity to the DNA binding domain, suggesting that ubiquitylation at K911 could disrupt DNA binding[28]. Interestingly, germline mutations in residues flanking K911 have been found in androgen insensitivity syndrome, indicating this area does make functionally important contacts. Together these findings suggest that K911 ubiquitylation may destabilize AR binding to chromatin. Finally, the K911R mutation has been reported in PCa, although just in one case of primary PCa[29]. This suggest that while mutation of this site can enhance AR activity in some contexts, lysine in this position may be important for AR function in PCa development or progression after androgen deprivation therapies.

Ubiquitylation at K313 has been identified previously, and mutation of this site to arginine was found to increase AR retention on chromatin but reduce AR transcriptional activity due to loss of p300 binding[11]. Notably, in addition to ubiquitylation at this site, we found methylation and acetylation, suggesting this site may be a regulatory hub. However, we found that mutation of this site to alanine did not have marked effects on AR stability or transcriptional activity. Interestingly, mutation of K313 combined with K318 (which we also found to be methylated) to alanine increased AR transcriptional activity. Importantly, these K313

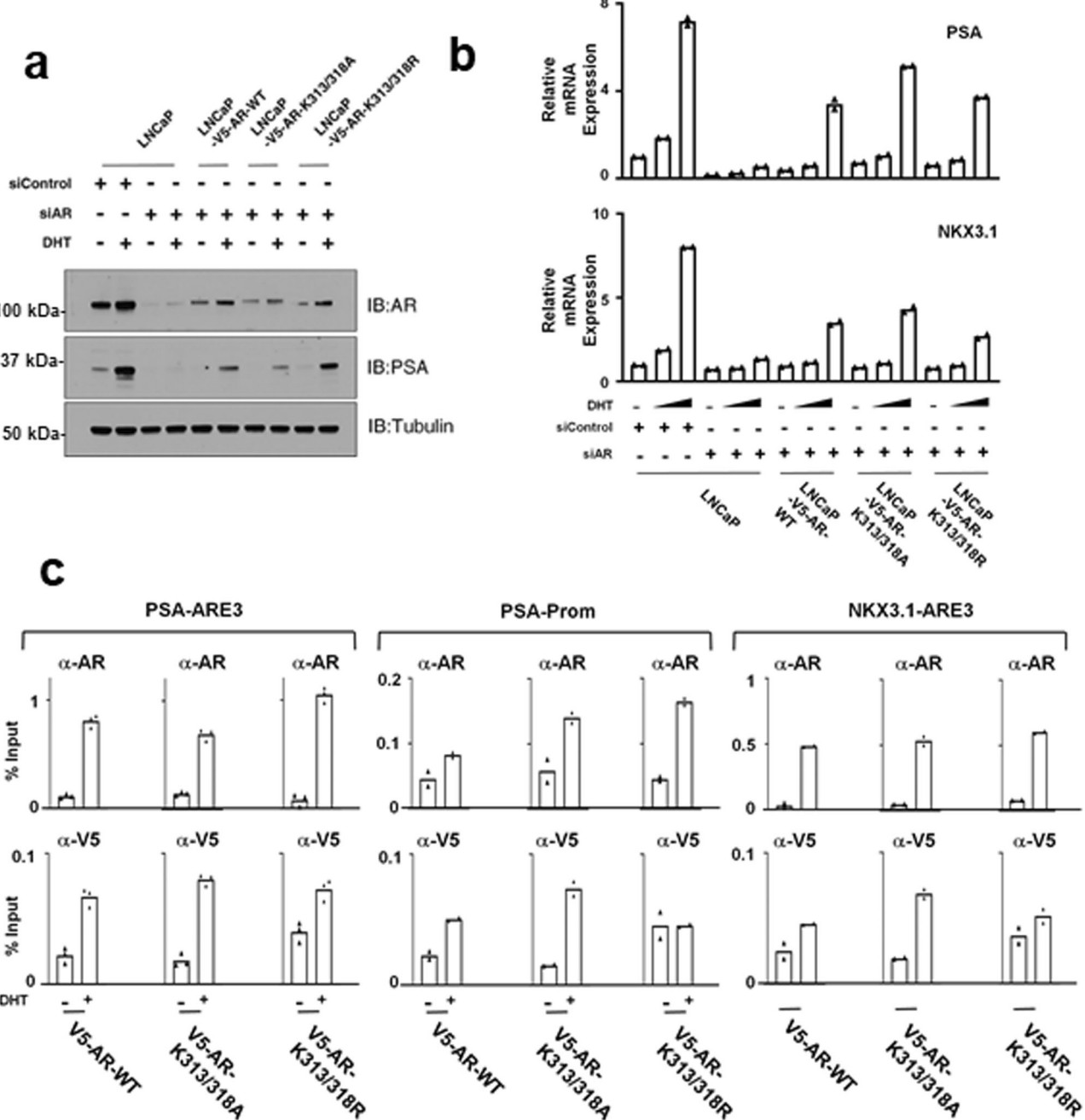

**Fig. 7 AR K313/318 mutations alter AR regulation of endogenous genes. a** LNCaP cells stably expressing wild-type or mutant ARs were treated for 2 days with an AR siRNA targeting the 5′ UTR (present only in the endogenous transcript), followed by 24 h in CSS medium with or without addition of DHT (10 nM). Lysates were then immunoblotted with an anti-AR or anti-PSA Ab. **b** LNCaP cells stably expressing wild-type or mutant ARs were treated for 2 days with an AR siRNA targeting the 5′ UTR, followed by 24 h in CSS medium with or without addition of DHT (1 nM or 10 nM). Levels of PSA and NKX3.1 mRNA were then determined by qRT-PCR. Data are mean of biological replicate samples. **c** LNCaP cells stably expressing wildtype or mutant ARs (V5 tagged) were cultured for 24 h in CSS medium and then stimulated with 10 nM DHT for 4 h. Recruitment of total AR or V5 tagged AR to the AREs in the PSA and NKX3.1 genes were then determined by ChIP-qPCR. Data are expressed as % of input DNA and mean of biological replicates are shown.

mutations would abrogate methylation and acetylation as well as ubiquitylation, and it is likely that the functional consequences of each PTM would be distinct. Therefore, further studies are needed to determine whether and how specific PTMs at this site modulate AR function. Interestingly, while we found that the K313/318 R mutant retained transcriptional activity, by ChIP it had increased basal chromatin binding and decreased fold induction by DHT, which is consistent with the previous study that examined a K313R mutant[11].

In addition to ubiquitylation, we identified multiple additional PTMs. Most of the serine and threonine phosphorylation sites have been reported previously, except S522. We also identified tyrosine phosphorylation at Y531 and Y621, which have not been reported previously, but their function remains to be determined. Interestingly, we found very frequent arginine methylation, with R20 methylation being most common. Several arginine methyltransferases have been found to enhance AR activity[30,31], likely through effects on AR associated proteins or possibly AR itself,

while PRMT5 has been reported to directly methylate AR at R761 and inhibit AR activity[32]. Further studies are needed to determine the function of arginine methylation and the corresponding arginine methyltransferases.

## Methods

**Reagents**. Antibodies were from the following sources: AR (Santa Cruz, Cat. sc-816), PSA (Meridian Life Science, Cat. K92110R), Flag-M2 (Sigma-Aldrich, Cat. F3165), anti-V5 (ThermoFisher, catalog no. R960-25), GAPDH (Abcam, Cat. Ab9485), Tubulin (EMD Millipore, Cat. MAB3408). Antibodies were used for immunoblotting at final concentration of 1 μg/ml. Molecular weights were determined by colored markers (BioRad) and their positons were marked off on the films by overlaying with the membranes. VLX1570 and b-AP15 were from Selleckchem. USP21 was kindly provided by D. Finley (Harvard Medical School). Cell fractionation was performed using a kit from ThermoFisher (Cat. 78840). The siRNAs were control siRNA (Dharmacon, Cat. D-001810-01-05) and AR siRNA (AR-1981, target sequence: AGGUUCUCUGCUAGACGACdTdT). The primers for ChIP-qPCR were: PSA-ARE3: Forward, 5-GCCTGGATCTGAGAG AGATATCATC-3; Reverse, 5-ACACCTTTTTTTTTCTGGAT TGTTG-3; PSA-promoter: Forward, 5-TCCTGAGTGCTGGT GTCTTAG-3; Reverse, 5-CAGGATGAAACAGAAACAGGG-3; NKX3.1-ARE3: Forward, 5'-CTGGCAAAGAGCATCTAGG G-3'; Reverse, 5'-GGCACTTCCTGAGCAAACTT-3'. RNA isolation was carried out using the TriZOL reagent (Ambion) and the qRT-PCR analysis on gene expression was performed with the TaqMan One-Step RTPCR Master Mix Reagents (Applied Biosystems). Primers were annealed at 55 °C for 30 s follow by extension at 72 °C for 60 s. The TaqMan primer-probe sets for PSA/NKX3.1 (FAM labeled) and the internal control GAPDH (VIC-TAMRA labeled) transcripts were purchased as inventoried mixes from Applied Biosystems.

**Cell lines and vectors**. Cell lines were obtained from ATCC. Identity was confirmed by STR testing, and they tested negative for Mycoplasma. Wild-type AR expression vectors with N-terminal Flag or V5 epitope tags were cloned into the Gateway entry vector (pDONR223-AR-WT) and then modified by site directed mutagenesis. Mutagenesis PCR was performed based on this construct to generate pDONR223 vector carrying AR-K911R, K837R, K826R, K823R and K639R. These AR-WT and AR lysine mutant cDNAs were subcloned into the lentiviral pLenti6.3-DEST-V5 vector and stable LNCaP and PC3 cell lines expressing these AR forms were established. The K313 and K318 mutant ARs were generated in the Flag M4 AR vector (p3XFLAG-CMV-10 backbone)[33]. pDONR223 vectors for SMYD1, SMYD2, KMT5A, SETD7, SET8 and LSD1 were purchased from the PlasmID Repository of Harvard Medical School. pLenti6.3-DEST-V5 vectors for expression of SMYD1, SMYD2, KMT5A, SETD7, SET8 and LSD1 were generated using the Gateway Technology with Clonase II (Invitrogen, catalog no. 12535-029).

**Coimmunoprecipitation and mass spectrometry**. Immunoprecipitated androgen receptor (AR) was run via SDS-PAGE and bands corresponding to unmodified AR (~110kD) and an area above ~125 kD were excised. Gel pieces containing protein were reduced with 55 mM DTT, alkylated with 10 mM iodoacetamide (Sigma-Aldrich), and digested overnight at 37 C with TPCK modified Trypsin/LysC (Promega) at pH = 8.3. Peptides were analyzed by microcapillary reversed-phase liquid chromatography-tandem mass spectrometry (LC-MS/MS) using a high resolution hybrid QExactive HF Orbitrap mass spectrometer (Thermo Fisher Scientific) via HCD with data-dependent analysis (DDA) with 1

MS1 scan followed by 8 MS2 scans per cycle (Top 8) using an EASY-nLC 1200 nanoflow UPLC (Thermo Fisher Scientific) at 300 nL/min using self-packed 15 cm length × 75 μm i.d. C18 fritted microcapillary columns. Solvent gradient conditions were 90 min from 3% B buffer to 38% B (B buffer: 100% acetonitrile; A buffer: 0.9% acetonitrile/0.1% formic acid/99.0% water). MS/MS spectra were analyzed using Mascot version 2.6 (Matrix Science) by searching the reversed and concatenated human protein database (http://www.ebi.ac.uk/uniprot/database/download.html) with a parent ion tolerance of 15 ppm and fragment ion tolerance of 0.05 Da. Carbamidomethylation of Cys was specified as a fixed modification and oxidation of Met, deamidation of Asp/Gln, phosphorylation of Ser/Thr/Tyr, ubiquitinylation of Lys, methylation and dimethylation of Lys/Arg and acetylation of Lys/Arg as variable modifications. Results were imported into Scaffold Q + S 5.0 software (Proteome Software, Inc.) with a peptide false discovery rate (FDR) of ~1.1%[34].

**Luciferase assay**. Cells were grown in DMEM medium containing 10% FBS. For androgen-starving conditions, cells were grown in medium containing 5% CDS. For transfection, cells were grown in normal growth medium to 80% confluence and plasmid DNAs and the ARE4-LUC reporter construct (pGL-ARE4-Luc) were transfected with LipofectAMINE 2000. After overnight transfection, the cultures were refreshed with indicated medium for treatments. Luciferase activities were measured with the Dual-Luc assay kit (Promega). The ratios between Firefly and Renilla luciferase activities were then determined.

**Statistics and reproducibility**. GraphPad Prism 9 Software (GraphPad Software Inc.) was used for all statistical analyses unless otherwise specified, and data points shown represent biological replicates. Blots shown are representative of at least three independent experiments.

**Reporting summary**. Further information on research design is available in the Nature Portfolio Reporting Summary linked to this article.

## Data availability

The source data for Figs. 5, 6, and 7, and Supplementary Figs. 5 and 6 is available in Supplementary Data 1. Uncropped gels are included in the Supplementary Figs. 7–18. Mass spectrometry data have been deposited in the MassIVE data base (https://massive.ucsd.edu/) with the accession code MSV000093051. Any further data not included in the text will be made available upon request.

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

## Acknowledgements

We thank Susanne Breitkopf for assistance with LC/MS/MS. This study was supported by Department of Defense (DoD) Prostate Cancer Research Program Postdoctoral Training Awards to Z.Y. (W81XWH-13-1-0254) and Y.G. (W81XWH-14-1-0245), DoD Prostate Cancer Research Program Idea Development Award to S.P.B/ (W81XWH-13-1-0266), DF/HCC-Prostate Cancer SPORE P50 CA090381 (S.P.B.), NIH P01 CA163227 (S.P.B.), NIH P01 CA120964 (J.M.A.), a Research Fellowship from Gunma University Hospital (S.A.), and Prostate Cancer Foundation Challenge Award (S.P.B.).

## Author contributions

The authors made the following contributions: conceptualization, S.A., Y.G., Z.Y., L.X.; methodology, S.A., Y.G., Z.Y., J.M.A.; formal analysis, S.A., Y.G., Z.Y., L.X., S.P.B.; investigation, S.A., Y.G., Z.Y., L.X., L.W., T.Z., S.C.; resources, S.P.B.; data curation, S.P.B.; writing original draft, S.P.B.; review and editing, S.A., Y.G., Z.Y., L.X., M.N.; supervision, S.P.B.; project administration, SPB; funding acquisition, S.P.B.

## Competing interests

The authors declare no competing interests.

## Additional information

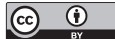

