## [Peer Review File · Communications Biology]

Reviewers' comments:

Reviewer #1 (Remarks to the Author):

The authors have performed extensive analysis to identify and characterize ubiquitination sites in the androgen receptor (AR). In addition to the ubiquitination, the authors have also performed mass spectrometric analysis to determine the other post translational modifications (PTM) in AR. The authors, in addition to previously determined PTMs, found new sites. One of the new sites found include K911. Mutagenesis studies unfortunately did not produce major functional changes in the AR. The authors have to be commended for conducting this extensive analysis. Though no major phenotype was identified with any of the sites, still the manuscript produces extensive information on AR biology, PTM sites, and ubiquitination. Considering that the current trend is to degrade the AR using PROTACs, identification of these sites is timely.

Some of the minor comments are provided below.

1. In figure 5B, the authors have not shown the effect of 1 nM DHT on FKBP5 with K911R.
2. The authors have not provided any explanation for not creating single mutations of K313R and K318R. They have created K->A mutations, but not K->R mutations.
3. The difference in gene expression with mutants shown in figure 7B is marginal.

Reviewer #2 (Remarks to the Author):

This manuscript is focused on an interesting and important topic, regulation androgen receptor (AR) stability. The investigators identify a new site of AR ubiquitylation, K911 and show that it increases AR stability, chromatin binding and transcriptional activity.

Major concern:

The topic of K313 and K318 modification should be better introduced. Further, almost all of the data regarding the impact of modification at these sites is negative (figures 6, 7 and 8) and should be combined into less figures or at least partially put into the supplemental data. The one piece of data showing that the DHT-driven expression of PSA and Nkx3.1 is greater with the K313/318A mutant than WT or the K313/K318R mutant (figure 7B) shows a tiny difference and should be better quantified. Perhaps a genomic approach would be more telling?

Minor concerns:

Page 4

"...while the series of lower molecular weight bands are consistent with sumoylation..."
Please explain why.

Fig. S1

Define GA.

Figure 4D

The blots should be quantified.

Reviewer #3 (Remarks to the Author):

The authors conducted a thorough evaluation of AR ubiquitylation sites, including a novel site at K911 and showed that the latter may increase AR stability, chromatin binding, and transcriptional activity. Overall, the study is worth publishing and expands the current knowledge on this very specific area.

Some points that need to be addressed are discussed below:

- Fig 3A bottom is too dense and the blot seems to be "cut" abruptly
- Fig5D H3K4Me lane is too dense
- it would be really interesting, to provide a clinical view as well of the study's findings in the discussion. E.g. How often is this novel site at K911 found mutated in most patient studies (e.g, from TCGA and others)? Do the authors speculate that patients with tumors harboring mutated AR at K911 would respond better/worse to novel AR-signaling pathway pharmacologic inhibitors including abiraterone, enzalutamide, apalutamide, darolutamide?

Response to reviewers' comments:

Reviewer #1 (Remarks to the Author):

The authors have performed extensive analysis to identify and characterize ubiquitination sites in the androgen receptor (AR). In addition to the ubiquitination, the authors have also performed mass spectrometric analysis to determine the other post translational modifications (PTM) in AR. The authors, in addition to previously determined PTMs, found new sites. One of the new sites found include K911. Mutagenesis studies unfortunately did not produce major functional changes in the AR.

The authors have to be commended for conducting this extensive analysis. Though no major phenotype was identified with any of the sites, still the manuscript produces extensive information on AR biology, PTM sites, and ubiquitination. Considering that the current trend is to degrade the AR using PROTACs, identification of these sites is timely.

Response: We thank the reviewer for their assessment that the work is timely.

Some of the minor comments are provided below.

1. In figure 5B, the authors have not shown the effect of 1 nM DHT on FKBP5 with K911R.

Response: This was inadvertently edited out, but has now but put back in the revised figure. It shows an increase relative to 1.0 nM with the WT.

2. The authors have not provided any explanation for not creating single mutations of K313R and K318R. They have created K->A mutations, but not K->R mutations.

Response: We created and examined cells with the double K to R mutations initially. As the effect on endogenous genes was modest (Fig. 7), we decided it would not be informative to go back and determine whether the effect was dependent primarily on one or the other site. We now clarify this point in the text.

3. The difference in gene expression with mutants shown in figure 7B is marginal.

Response: As above, we agree it is modest, with the K to A mutations causing an ~50% increase for PSA, but only ~20% for NKX3.1. We now point out in the revised text that the increase is modest.

Reviewer #2 (Remarks to the Author):

This manuscript is focused on an interesting and important topic, regulation androgen receptor (AR) stability. The investigators identify a new site of AR ubiquitylation, K911 and show that it increases AR stability, chromatin binding and transcriptional activity.

Major concern:

The topic of K313 and K318 modification should be better introduced.

Response: We have provided more background on the K313 and K318 sites in the Introduction and Results sections.

Further, almost all of the data regarding the impact of modification at these sites is negative (figures 6, 7 and 8) and should be combined into less figures or at least partially put into the supplemental data.

Response: The modifications do have a modest stimulatory effect, which is in contrast to findings in a previous study, and hence indicate that ubiquitylation at this site is not generally required for transcriptional activity. Therefore, we would argue that the data are not entirely negative. Nonetheless, we would agree that figure 8 could be viewed as negative data, and have moved this to the supplementary figures.

The one piece of data showing that the DHT-driven expression of PSA and Nkx3.1 is greater with the K313/318A mutant than WT or the K313/K318R mutant (figure 7B) shows a tiny difference and should be better quantified. Perhaps a genomic approach would be more telling?

Response: We agree that the difference is modest, but it is consistent with the increased activity in the reporter gene assays in figure 6. We now clarify in the text that the effect is modest and indicate that it is ~50% on PSA and ~20% for NKX3.1 for the K to A mutants. We agree that further genomic approaches such as knocking in the mutation and doing in vivo functional studies may be more telling, but that is beyond the scope of the current study.

Minor concerns:

Page 4 "...while the series of lower molecular weight bands are consistent with sumoylation..."
Please explain why.

Response: The wording of that sentence was confusing. The gels show a series of discrete AR bands running about 20-60 kD above AR which do not appear to be ubiquitylated. This would be consistent with sumoylation as each SUMO addition increases apparent molecular weight by about 20 kD on SDS-PAGE gels. This now clarified in the revised text on page 4.

Fig. S1 Define GA.

Response: GA is geldanamycin. This has been added to the text and figure legend.

Figure 4D. The blots should be quantified.

Response: We now show quantification of these bands relative to the control in the revised figure.

Reviewer #3 (Remarks to the Author):

The authors conducted a thorough evaluation of AR ubiquitylation sites, including a novel site at K911 and showed that the latter may increase AR stability, chromatin binding, and transcriptional activity. Overall, the study is worth publishing and expands the current knowledge on this very specific area. Some points that need to be addressed are discussed below:

Fig 3A bottom is too dense and the blot seems to be "cut" abruptly

Response: The dense bottom band at ~75 kD is AR-V7, and the figure shows a short exposure (SE) of this area right above the long exposure. The blot similarly shows a short exposure (SE) to the full length AR band at ~110 kD. We now clarify this in the figure legend, and have also separated the AR and AR-V7 blots to avoid confusion.

Fig5D H3K4Me lane is too dense

Response: We agree and have replace it with a shorter exposure of the gel.

It would be really interesting, to provide a clinical view as well of the study's findings in the discussion. E.g. How often is this novel site at K911 found mutated in most patient studies (e.g, from TCGA and others)?

Response: We note in the discussion that the K911 mutation has been reported in PCa (reference 29). However, this was just one case and it has not been found in further cases in public domain data sets. This is now clarified in the revised Discussion. We also now point out that the absence of recurrent mutations at this site in PCa could possibly reflect an important function for this lysine. We note that germline mutations around this lysine are causes of androgen insensitivity syndrome, suggesting that this area may have important novel functions. Hence, although we find that mutation of K911 can enhance AR transcriptional activity in our assays, it may impair some critical in vivo functions in tumor cells.

Do the authors speculate that patients with tumors harboring mutated AR at K911 would respond better/worse to novel AR-signaling pathway pharmacologic inhibitors including abiraterone, enzalutamide, apalutamide, darolutamide?

Response: A straightforward guess would be that they would respond worse as this AR is somewhat hyperactive. However, if it emerged in men treated with just a GnRH agonist, it may be a biomarker of tumors that are highly AR dependent. In this case these men might have better responses to these more potent AR targeted drugs.

REVIEWERS' COMMENTS:

Reviewer #1 (Remarks to the Author):

None. The authors have addressed my comments.

Reviewer #2 (Remarks to the Author):

This revision addresses my previous concerns and is suitable for publication in Communications Biology.